# Removal of Volatile Organic Compounds (VOCs) from Air: Focus on Biotrickling Filtration and Process Modeling

Piotr Rybarczyk

Department of Process Engineering and Chemical Technology, Faculty of Chemistry, Gdańsk University of Technology, Narutowicza 11/12 Street, 80-233 Gdańsk, Poland; piotr.rybarczyk@pg.edu.pl; Tel.: +48-58-347-12-29

**Abstract:** Biotrickling filtration is a well-established technology for the treatment of air polluted with odorous and volatile organic compounds (VOCs). Besides dozens of successful industrial applications of this technology, there are still gaps in a full understanding and description of the mechanisms of biotrickling filtration. This review focuses on recent research results on biotrickling filtration of air polluted with single and multiple VOCs, as well as process modeling. The modeling offers optimization of a process design and performance, as well as allows deeper understanding of process mechanisms. An overview of the developments of models describing biotrickling filtration and conventional biofiltration, as primarily developed and in many aspects through similar processes, is presented in this paper.

**Keywords:** biofiltration; biotrickling filtration; volatile organic compounds; process modeling; removal efficiency

## 1. Air Pollution with VOCs: Problem and Treatment Strategies

Air pollution with volatile organic compounds, such as hydrocarbons, ethers, esters or alcohols that resultfrom both anthropogenic and industrial activities, poses a broad scope of health hazards to living organisms [1]. Volatile organic compounds easily evaporate, even at room temperature. Because these are chemicals with different chemical properties, including small molecular size with no electric charge, variable lipophilicity, and volatility, inhalation is the major route of their absorption by living organisms [2]. Inhalation-related hazards are not only related to short-term exposure and possible feelings of discomfort, but mainly to long-term exposure and possible unknown health effects. This has also become evident from the perspective of COVID-19 epidemics [3]. For example, number of reported infections and death cases related to COVID-19 may be related to the severity of atmospheric pollution in a given territory [4]. It is obvious, therefore, that efficient, reliable, and economic methods of air treatment need to be developed and improved.

Various treatment methods of air polluted with VOCs have been established worldwide. These methods include combustion [5], absorption [6], adsorption [7], condensation [8,9], membrane processes [10], oxidative catalysis [11,12], UV-oxidation [13], and non-thermal plasma processes [14]. Besides the fact that such gas treatment methods have plenty of advantages, like rapid start-up periods and efficient removal rates of the pollutants, they are usually costly, especially in operation, and may consume considerable amounts of chemicals that generate hazardous by-products. Thus, some of these methods may be considered unsustainable. Due to high efficiency, their possibility to economically treat large volumes of gases, their limited or nonexistent use of chemicals, and their capability of operating at ambient temperatures (10–40 °C) and under atmospheric pressure, biological methods find promising applications as deodorization methods [15–18]. These methods mainly include biofiltration, bioscrubbing, biotrickling filtration, and other bioreactor configurations, and sometimes include combinations of these methods with other physico-chemical methods [19–22].

Biological methods of VOC removal from air have been known for about one hundred years, and constitute the prevailing share of conventional biofiltration processes. Initially, microbial reactions in soil were employed for air treatment [18,23]. Biofiltration finds applications in odor control for different waste gas streams, e.g., in rendering plants, chemical factories, agriculture, and animals farming. Biofilters, i.e., bioreactors in which biofiltration is carried out, are fixed-bed reactors that are inhabited by active microorganisms. These microorganisms are able to degrade VOCs present in air, and use them as carbon and energy sources. Biofiltration mechanisms of VOC removal include a mix of various phenomena, like absorption, adsorption, diffusion, and biodegradation. As a result of biofiltration, VOCs are converted into biomass, $CO_2$, and water [18,24]. A distinctive feature of biofiltration processes, also termed conventional biofiltration (abbreviated BF), is the lack of mobile liquid phases inside the biofilter. BF can be realized in open or closed bioreactors.

This paper focuses on biotrickling filtration (abbreviated BTF). According to a definition by Deshuses, Gabriel, and Cox [25,26] biotrickling filters differ from conventional biofilters in two main areas i.e., the packing type and liquid phase occurrence. Biotrickling filters are packed with inert or manufactured media, over which a liquid phase is trickled; thus, biotrickling filters are by definition bioreactors that contain gas, solid and liquid phases. This is contrary to conventional biofilters, which are packed with organic materials, are naturally inhabited by various types of microorganisms, and which humidify the polluted air in a separate chamber prior to entering the biofilter [27].

This paper aims to provide an update for the removal of VOCs using biological methods, with a focus on biotrickling filtration research on the removal of single VOCs and multiple VOCs, as well as a review of the modeling of biotrickling filtration as a core of this work. An overview of conventional biofiltration modeling is also presented as a basis for the development of BTF modeling. To the best knowledge of the author, no paper has yet presented a comprehensive outlook on the mentioned aspects of biotrickling filtration up to this point.

## 2. Review Methodology

This review was prepared on the basis of the literature research using well-established scientific databases (e.g., Science Direct) with over 250 references, out of which 174 were finally referred to. The references on the biotrickling filtration research were limited to the last ten years in order to present the most updated results. The exception is the review of treating multiple VOCs, for which the timespan is from 2004 to 2022. The review of the development of models for biofilters and biotrickling filters covered the period from about 1990 to 2022, to show the broadest possible perspective on BF and BTF modeling aspects. The paper is focused on biotrickling filtration; however, attention is also paid to conventional biofiltration as primarily developed and in many similar aspects to BTF.

## 3. Biotrickling Filtration for the Removal of VOCs from Air: Background, Current Problems, and Research Overview

Biotrickling filtration is a biotreatment method that enables the removal of gaseous contaminants from the air [16,23,28]. The process consists of passing a polluted gas through a packed layer. Elements of this layer are usually inert. Prior to starting the process, these elements are inoculated with either a single microorganism (usually for model investigations or for testing a new species with high expected potential to degrade VOCs) or a mixed microbial inoculum (a consortium of selected microbial species or an inoculum originated from the activated sludge of a wastewater treatment plant). As a result of supplying the biotrickling filter with polluted air (containing oxygen as well as pollutants, which serve carbon sources for microbes), as well as spraying the packed layer with a so-called trickling liquid, a biofilm is formed over the elements of a packed layer [16,29]. Within this biofilm, a biodegradation of selected/all air pollutants is possible, which results in the emission of cleaned air from the bioreactor. When the biofiltration process is realized effectively, the treated air leaving the biotrickling filter contains only a fraction or practically no VOCs,

but contains slightly elevated $CO_2$ concentrations, due to biodegradation, when compared to the inlet stream of air. When it comes to the liquid phase i.e., the trickling solution, it is usually an aqueous solution of mineral salts and micronutrients, e.g., vitamins. When used in the biotrickling filtration process, the liquid absorbs by-products of biodegradation, flushes fractions of dead microbial matter, and may contain variable concentrations of VOCs, depending on their solubility and biodegradation rate. The trickling liquid is usually recirculated unless a certain level of its evaporation or absorption/intake of the above-mentioned substances is reached. Then, the liquid must be regenerated e.g., by either removing and discarding the remaining volume and introducing a fresh liquid, or by daily liquid make-ups to maintain a fixed level.

There are recent review papers on controlling air pollution using biotrickling filters [16,23,30] that also underlie the possibility of using the biotrickling filtration of VOCs in wastewaters treatment [31], in which the process parameters and their effects on the biofiltration mechanisms are thoroughly discussed. Another excellent recent review paper in the field of biotrickling filtration deals with the effects of the liquid phase on the gas contaminant removal [32], as well as on the enhanced removal of hydrophobic air contaminants as a result of the addition of a surfactant or a hydrophilic gas-phase component [28]. An interesting review paper [33] concerns mass transfer aspects for biofilters and biotrickling filters, among others. The paper discusses the main mechanisms for the resistance of component transfers from gas to biofilm phases, which mainly include mass transfer limitations or kinetic limitations. The paper discusses the methods of determining mass transfer rates, as well as the modes of influencing and enhancing the mass transfer properties. These aspects are especially important in the case of modeling any process occurring as a result of contact between at least two different phases. Yang et al. [34] provided an interesting paper on the possible interactions between VOCs when multiple gas pollutants were treated in biofilters. A recent review paper by Danila et al. [35] discusses the issues of packing humidity and irrigation patterns for biofilters, which is an important, though seldom discussed, process parameter. In a recent review paper by Marycz et al. [36] the advantages of fungi utilization in biotrickling filters are discussed.

Interestingly, biotrickling filters for industrial applications can be obtained by retrofitting the existing chemical scrubbers, as discussed by Gabriel and Deshusses [37].

The conditions for conducting the biotrickling filtration process are mainly determined by the volumetric flow rate of treated gas ($Q$, $m^3$), the concentration of VOCs in the inlet gas stream ($C_{in}$, g m$^{-3}$ or *ppmv*), the biofilter dimension (cross-surface area represented by internal diameter, $d_{in}$, and height of packing, $H$), the resulting effective volume of the biofilter packing ($V$, $m^3$), and the packing characteristics (porosity, surface area) [16,23,30]. These parameters are usually represented by the values of specific inlet loading (*IL*, g m$^{-3}$ h$^{-1}$) and empty bed residence time (*EBRT, s*). Process efficiency is usually reported using the values of removal efficiency (*RE*, *dimensionless* or %) and elimination capacity (*EC*, g m$^{-3}$ h$^{-1}$). While both RE and EC are dependent upon the inlet concentrations of VOCs in the gas phase, EC gives a clear and comparable value representing the actual performance capacity of a given biofilter (what mass of VOCs can be removed using a given volume of biofilter per unit of time). Additionally, the value of EBRT is also an important parameter for the comparison of different processes, especially from a practical viewpoint. However, when referring to the process efficiency with the use of RE, one should always present the inlet VOC concentrations, because sometimes a high RE can be reached only for low $C_{in}$ or IL values, and thus may be misleading when evaluating biofiltration performance. The formulae for calculating the above-mentioned parameters are given below.

$$IL = Q{\cdot}C_{in}{\cdot}V^{-1}, \tag{1}$$

$$EBRT = V{\cdot}Q^{-1}, \tag{2}$$

$$RE = (C_{in} - C_{out}){\cdot}C_{in}^{-1}, \tag{3}$$

$$EC = Q{\cdot}(C_{in} - C_{out}){\cdot}V^{-1}, \tag{4}$$

where $C_{out}$ is the VOC concentration in the outlet gas stream (g m$^{-3}$).

For a given biotrickling filter set-up, including microorganisms and fixed process conditions (i.e., VOC concentrations in the gas phase and its flow rate), biotrickling filtration performance can be regulated by manipulating the trickling liquid velocity and frequency, the process temperature, the pH and composition of the trickling liquid, the packing material, and the feeding and biomass growth control strategies. However, it should be taken into account that the frequency of the trickling process cannot be implemented in all applications, as agricultural applications strongly require continuous irrigation. A brief summary of the effects of these parameters on biofiltration performance is given in Table 1.

**Table 1.** Selected operating and process parameters and their effects of the performance of biotrickling filters treating air polluted with VOCs.

| Parameter | Effects | References |
|---|---|---|
| Liquid trickling velocity and frequency | The mode (intermittent/continuous) and velocity of trickling affects the humidity of the packing (important for biomass growth), ensures a supply of minerals/microelements for microbial growth, rinses out biodegradation by-products and dead matter, and plays role in managing the pressure drop across the packing; increased trickling velocity may result in decreased mass transfer of the hydrophobic VOCs, but on the other hand may enhance the mass transfer of hydrophilic VOCs and result in more uniform water distribution throughout the biofilter bed | [32,38] |
| pH, temperature and composition of trickling liquid | Trickling liquid can buffer the pH in the optimal range for microbial growth (e.g., fungi prefer an acidic pH); the modification of trickling liquid composition using, e.g., surfactants (also biosurfactants produced in situ by microbial metabolism) or silicone oil, can enhance the removal of hydrophobic VOCs from air; the increase in process temperature may result in increased process efficiency, but also increased pressure drops across the packing layer | [28,39,40] |
| Packing material | Depending on the porosity and surface area, packing materials define the available surface for biofilm development and growth, and consequently affect the interfacial area for mass transfer from the gas to biofilm/liquid phase; this parameter greatly affects the pressure drop in the biofilter (low porosity and high surface area may lead to fast clogging of the bed); Bio-based packing materials seem to be promising in the development of biofiltration methods for hydrophobic VOCs, e.g., those coming from the pharmaceutical industry. | [30,32,41,42] |
| Feeding strategy | Continuous, intermittent, or periodic shut-down of the polluted gas supply (e.g., for weekends); the mixing of gas streams from two sources with distinctly different waste gas compositions (e.g., hydrophilic and hydrophobic components) | [43,44] |
| Biomass growth control strategy | The control of a pressure drop and avoidance of the bed clogging, e.g., by periodic switching of the co-/counter-current gas-liquid flow patterns, or compressing and squeezing out of the packing (bed clogging results in the deterioration of biofilter performance, and sometimes the recovery is not possible); mechanical, chemical, or biological (e.g., predators) methods are used | [45,46] |

The main body of evidence of research on biotrickling filtration covers the treatment of single air pollutants. Such an approach allows for the definition of process mechanisms with respect to particular compounds, and may be profitable for practical application when a single compound is dominating in the gas mixture. Table 2 presents selected research results on the biotrickling filtration of single VOCs with a focus on the enhancement of hydrophobic VOC biotrickling filtration, as well as presents advances in process performance, design, and construction modifications.

However, in the industrial practice and in various processes of municipal waste stream treatments, including wastewaters and solid waste processing, mixtures of various VOCs are present. The co-existence of a number of compounds to be treated simultaneously affects the process mechanism and performance, and requires a different design approaches than for single compounds. Additionally, the process modeling requires a different assumptions and approaches; e.g., Baltzis et al. [47] considered the case that different compounds underwent biodegradation carried out by different biofilm types, i.e., different types of microorganisms. Table 3 presents selected resent research on the biotrickling filtration of multiple VOCs. The data was selected from the perspective of the evaluation of process mechanisms for the effects of substrate interactions (e.g., co-metabolism and various types of synergistic effects) on the process performance.

Besides considerable research on the biotrickling filtration of VOC mixtures and the increasing number of scientific papers being published on the topic [48], the interactions between the mixed pollutants and their effects on the biofiltration performance and mechanisms still need to be explored [49–52].

For example, when the co-treatment of multiple VOCs takes place in a biotrickling filter, several aspects of the process mechanism may lead to either improved or decreased process efficiency [34]. The decrease in biofiltration performance may be a result of substrate inhibition or toxic effects. For example, methanol at low concentrations may promote the removal of hexane, while at higher concentrations may lead to the decreased removal of hexane, because it becomes the dominant carbon source and can become toxic to microorganisms [50]. An increase of BTF performance may, on the other hand, result from improving the mass transfer of the hydrophobic substrates as a result of the increased solubility in the more hydrophilic compounds, which thus enhances biodegradation and leads to increased microbial growth and the promotion of co-substrate degradation [49]. Some VOCs, like toluene, can also modify the production and composition of extracellular polymeric substances (abbreviated as EPS) [40,51]. EPS are mainly composed of proteins and sugars, and the relative ratio of these can affect the biofilm hydrophobicity i.e., biofilm hydrophobicity increases with the increased ration of proteins to sugars. It is evident that the higher hydrophobicity of biofilm contributes to the mass transfer of hydrophobic VOCs from gas to biofilm, thus resulting in their enhanced biodegradation. It should also be mentioned that the degradation of hydrophobic VOCs can be accelerated by the use of packing materials with enhanced adsorptive capacities, i.e., hydrophobic VOCs can adsorb and accumulate on these carriers, and mass transfer to microbial cells and final biodegradation is enhanced by higher VOC gradients between the carrier and the biofilm [53–56].

Research results presented in Table 2 show a growing interest in the investigations of biofiltration performance for gas mixtures that mimic the real waste gas compositions. Various configurations and process conditions have been studied from the laboratory pilot scale of a BTF. A considerable problem during the design and operation of biotrickling filters is their limited efficiency in treating hydrophobic volatile organic compounds [16,28]. This problem is mainly related to the low solubility of such compounds in the aqueous solutions, thus resulting in their low availability for microorganisms, as well as their usually high toxicity and resistance to biodegradation. Thus, modes of enhancing the removal of hydrophobic VOCs are of special interest nowadays, and present promising results when the utilization of two-phase biotrickling filters is considered [57].

The overview of current research on biotrickling filtration (Tables 2 and 3) includes selected process conditions and parameters, like bioreactor dimensions, packing type,

inoculum, process performance, and important outcomes for selected treated VOCs. Due to the dominance of papers discussing the removal of single air pollutants (Table 2), the selected recent research results from the period from 2011 to 2022 are revised. Due to a lower number of papers on investigations of the biofiltration of mixed VOCs, Table 3 presents selected results from a longer time perspective, i.e., 2004–2022. The results indicate that research has been mainly performed on laboratory-scale biofilters (with packing volumes of about several liters); however, there are also reports on pilot and full-scale applications. Dominant packing materials include ceramics, minerals, and polymeric materials, such as polyurethane foam, while bio-based materials, structured packing, or other engineered and novel materials are in minority. The most common inocula originate from activated sludge from wastewater treatment plants; however, research has also been conducted on the identification and testing of new microbial species that are effective in degrading VOCs. The majority of EBRT values range between 30-100 s, while the total VOCs concentrations in the inlet gas streams, irrespective of the number of compounds, range from about several tens to thousands of mg m$^{-3}$, which are approximately equivalent to hundreds of ppm (*v/v*). Current research goals include the improvement of biotrickling filtration performance for hydrophobic VOCs by using surface active substances, modified liquid phases, two-phase bioreactors, the addition of hydrophilic compounds, or the application of fungi. Other areas of development include scaling up processes, the management of secondary waste, and the incorporation of pre- or post-treatment processes to increase process efficiency or obtain broader utilitarian purposes, such as coupling technologies to cope with environmental issues. Another novel approach includes the design of biotrickling filters as microbial fuel cells for energy generation during waste air treatment [58–62].

**Table 2.** Selected recent research on biotrickling filtration of single VOCs.

| VOC | Bioreactor Dimensions: $d_{in} \times H$ [m]; V [dm$^3$] | Packing Material | Inoculum | Process Parameters | Interesting Outcomes | Reference |
|---|---|---|---|---|---|---|
| n-Butanol | 0.08 × 0.68; 2.5 | Peat & perlite, ceramic Raschig rings | No additional inoculation | EBRT: 60 s, $C_{in}$: 100–800 ppm | The extent of reduction in VOCs concentration in gas phase is not quantitatively equivalent to odor reduction | [63] |
| Triethylamine | 0.21 × 1; 52 | Lava rocks | Sludge from WWTP | EBRT: 31–312 s $C_{in}$: 600 mg m$^{-3}$ | Increasing of the gas flow rate is more cost-effective than increasing EBRT | [64] |
| Trimethylamine | 0.077 × 1.7; 5.6 | Polyurethane rings | *Aminobacter aminovorans* | EBRT: 85, 170 s IL: 0.2–12 g·m$^{-3}$·h$^{-1}$ | H$_2$S only slightly affected the removal efficiency of trimethylamine | [65] |
| Ethyl acetate | 0.2 × 0.14 (two sections) | Anode materials: carbon coke or carbide porous ceramic rings (CPCR) | Activated sludge from WWTP | EBRT: 60–120 s $C_{in}$: 0.54–3.23 g m$^{-3}$ | The use of CPCR resulted in higher removal efficiency of ethyl acetate that carbon coke | [61] |
| Methyl acrylate | 0.12 × 1.1; 6.44 | Ceramic particles | Activated sludge from WWTP | EBRT: 200–400 s $C_{in}$: 120–7505 mg m$^{-3}$ | Three-layer BTF was applied and it was the 1$^{st}$ packing layer that mainly reduced methyl acrylate concentrations in air | [1] |

**Table 2.** *Cont.*

| VOC | Bioreactor Dimensions: $d_{in} \times H$ [m]; V [dm$^3$] | Packing Material | Inoculum | Process Parameters | Interesting Outcomes | Reference |
|---|---|---|---|---|---|---|
| Chloroform | $0.076 \times 1.3$; 5.89 | Celite (pelletized diatomaceous earth) | Filamentous fungi | EBRT: 344 s $C_{in} = 5$ ppm | The application of ethanol as a co-metabolite; acidic conditions enhanced the fungi growth | [66] |
| | $0.076 \times 1.3$; 5.89 | Pelletized diatomaceous earth | Bacteria from WWTP | EBRT: 344 s $C_{in}$: 200 ppm | Aerobic and anaerobic conditions (with methanogenic bacteria) were investigated | [67] |
| Ethylbenzene | $0.1 \times 0.78$; 6.1 | Polyurethane sponge | Activated sludge from WWTP | EBRT: 30 s $C_{in}$: 2.5 g m$^{-3}$ | The addition of saponins to the liquid phase resulted in enhanced removal of ethylbenzene | [52] |
| | $0.1 \times 0.8$ (2 sections); 12.56 | Polyurethane sponge | Fresh biological from WWTP | EBRT: 30 s $C_{in}$: 1 g m$^{-3}$ | The high removal of ethylbenzene due to addition of biosurfactant (from piggery wastewater) to liquid phase | [68] |
| Styrene | $0.144 \times 1.63$; 20 | Polypropylene Ralu rings | *Pseudomonas sp.* E-93486 | EBRT: 62 s IL: 1 g m$^{-3}$ h$^{-1}$ | A three-phase dynamic mathematical model was proposed | [69] |
| | Six rectangular sections with total packing volume of 50 dm$^3$ | Fern or plastic chips | Activated sludge from WWTP | EBRT: 21 s $C_{in}$: 2.39 g m$^{-3}$ | The study provides a comparison of packing materials, presenting the superior performance of plastic over fern chips during styrene removal from air | [70] |
| | $0.08 \times 1.5$; 3.77 | Ceramic Raschig rings | Activated sludge from cooking plant | EBRT: 68 s IL = 180 g m$^{-3}$ h$^{-1}$ | The removal of styrene from air is enhanced by the acclimatization of a biotrickling filter to toluene-styrene mixture | [71] |
| Toluene | $0.12 \times 0.78$; 4.27 | Ceramic pellets | Fungi from activated sludge (e.g., *Fusarium*) | EBRT: 55 s IL = 100.3 g m$^{-3}$ h$^{-1}$ | A fungi-based biotrickling filter was applied to efficiently remove toluene from air | [72] |
| | $0.1 \times 0.6$ (approx.); 4.56 | Ceramsite particles | Fungi (*Fusarium*) | EBRT: 59 s $C_{in}$: 1.053 g m$^{-3}$ | The effects of bed porosity on pressure drop and its maintenance during biotrickling filtration process were studied | [73] |
| | Differential biotrickling filter with a packing volume of 0.4 dm$^3$ | Glass beads | Inoculum from previously working biotrickling filter | EBRT: 28 s IL: 472 g m$^{-3}$ h$^{-1}$ | A differential biotrickling filter was proposed in opposition to popular integral-column bioreactors | [74] |
| Cyclohexane | $0.318 \times 1$; 52 | Polyurethane foam | *Acidovorax sp.* CHX 100 | EBRT: 37 s $C_{in}$: 720 mg dm$^{-3}$ | A proposal for the removal of cyclohexane from air in a biotrickling filter using *Acidovorax sp.* | [75] |

**Table 2.** *Cont.*

| VOC | Bioreactor Dimensions: $d_{in} \times H$ [m]; V [dm$^3$] | Packing Material | Inoculum | Process Parameters | Interesting Outcomes | Reference |
|---|---|---|---|---|---|---|
| Hexane | 0.07 × 1; 2.4 | Perlite | *Fusarium solani* | EBRT: 78 s $C_{in}$: 7400 mg dm$^{-3}$ | The addition of different carbon sources to the trickling liquid may shorten the start-up period | [76] |
| | 0.076 × 1.3; 2.7 | Pellets of diatomaceous earth (Celite) | Fungi with dominant *Cladosporium* and *Rhodotorula* species (genus) | EBRT: 120 s $C_{in}$: 125 ppmv | The effects of pH and methanol addition on the removal of hexane were studied; low pH was favorable for enhanced removal of hexane | [77] |
| | 0.1 × 0.65; 5.1 | Reticular polyurethane sponge | Filamentous bacteria | EBRT: 30 s IL: 124 g m$^{-3}$ h$^{-1}$ | The intermittent mode of spraying and reticulated packing configuration, enhancing the mass transfer, enabled high removal of hexane | [78] |

**Table 3.** Selected research on biotrickling filtration of VOCs mixtures.

| VOCs Mixture | Bioreactor Dimensions ($d_{in}$ x H; V) | Packing Material | Inoculum | Process Parameters | Outcomes | Reference |
|---|---|---|---|---|---|---|
| Acetone, toluene, trichloroethylene | 0.07 × 0.6; 1.9 (bioreactor organized in 2 segments) | Granular activated carbon | Acclimated microbial seeds from industrial WWTP | EBRT: 155 s $C_{in}$ (of each VOC): 10–800 ppmv | Relative concentrations of VOCs in treated gas mixture as a crucial parameter for the functioning of microbes during biofiltration | [79] |
| Trichloroethylene, perchloroethylene | 0.07 × 0.6; 3 | Granular activated carbon | Mixed microbial consortium from soil and activated sludge | EBRT: 6–36s $C_{in}$: 35.4 ppmv (PCE) and 46.7 ppmv (TCE) | The application of photooxidation enhanced the removal of target pollutants in biotrickling filter, and various removal mechanisms in a biofilter were identified (hydrolysis, adsorption, biodegradation) | [80] |
| Hydrogen sulfide, methyl mercaptan, ammonia, VOCs mixture, dimethyl sulfide | 3.2 × 13 | OdourTeQ vessels | *Thiobacillus* for H$_2$S removal, heterotrophic microbes for VOCs removal | EBRT: 17 s | Industrial application | [81] |
| Methanol, ethanol, acetone, toluene, chloroform | 0.054 × 1; 1.7 | Polyvinyl chloride particles | Microbial consortium taken from previously operating bioreactors | EBRT: 25–68.6 s $C_{in}$: 1–4 g m$^{-3}$ (VOCs); 0.05–1 g m$^{-3}$ (chloroform) | High removal of target air pollutants and resistance to intermittent loadings | [82] |
| Methyl mercaptan, toluene, alpha-pinene and hexane | 0.08 × 1; 4 | Polyurethane foam cubes | Acclimated activated sludge | RE up to about 99% for hydrophilic and about 80% for hydrophobic VOCs at inlet concentrations of order of mg m$^{-3}$ for EBRT = 7 s | The addition of silicone oil enhanced and stabilized biofiltration performance; high overall mass transfer coefficients reported for polyurethane foam | [83] |

**Table 3.** *Cont.*

| VOCs Mixture | Bioreactor Dimensions ($d_{in}$ x H; V) | Packing Material | Inoculum | Process Parameters | Outcomes | Reference |
|---|---|---|---|---|---|---|
| Ethyl acetate, toluene, ethyl benzene, xylene, ethyl toluene, trimethylbenzene | $2 \times 2 \times 4$ (rectangular bioreactor); 6 m$^3$ | Ceramic particles, hollow plastic balls, Raschig rings | Enriched activated sludge from WWTP | EBRT: 7.2 s $C_{in}$: up to 150 mg m$^{-3}$ (sum of VOCs) | BTF combined with photocatalytic oxidation revealed high performance for VOC removal from air | [84] |
| Acetone, methyl ethyl ketone (MEK), toluene, styrene | Field-scale stainless steel BTF: $1.5 \times 1.5 \times 1.8$ [m] | Pall rings | Activated sludge from the local paint and coating plant wastewater treatment plant | EBRT: 14 s Total inlet loading of VOCs mixture: 30.5 m m$^{-3}$ h$^{-1}$ RE from 67 to 99 % | Mixed VOCs from paint and coating plant; higher removal of acetone and MEK than toluene and styrene; shutdown periods diminished BTF performance, especially for hydrophobic VOCs | [85] |
| Methanol, n-hexane | $0.076 \times 1.3$; 2.72 | Pelletized diatomaceous earth (Celite) | Aerobic microbial culture from secondary clarifier of an activated sludge system at Cincinnati municipal wastewater treatment plant | EBRT: 120 s RE of methanol about 99% regardless process conditions; RE of n-hexane decreased from about 98% to 60% when inlet loading increased from 1 to 13.2 g m$^{-3}$ h$^{-1}$ | Methanol significantly increased the removal of hexane in the BTF; switching of flow directions (co- and counter-current) was applied to ensure uniform biofilm formation across the packing; decrease in BTF performance was observed after bed backwashing due to high biomass loss; high methanol concentrations may have inhibited n-hexane biodegradation, as it is a more accessible carbon source | [50] |
| Hydrogen sulfide, methanol, α-pinene | $0.094 \times 0.7$; 4.55 | Polypropylene pall rings | Autotrophic H$_2$S-degrading culture and pure strains of *Candida boidinii*, *Rhodococcus erythropolis*, and *Ophiostoma stenoceras* | EBRT: 26 s Maximum elimination capacities of methanol, α-pinene and H$_2$S were 302, 175, and 191 g m$^{-3}$ h$^{-1}$, respectively | The α-pinene degraders developed more slowly than degraders for methanol and H$_2$S; diversified microbial population within the filter bed offered promising VOCs removal for industrial applications | [57] |
| Butanone, toluene, α-pinene, hexane | $0.08 \times 1$; 4 | Kaldness K1 rings | Activated sludge from Valladolid WWTP | EBRT: 6 s RE for inlet concentrations of butanone (3 mg m$^{-3}$); toluene (1.4 mg m$^{-3}$); α-pinene (1.4 mg m$^{-3}$) and hexane (1.3 mg m$^{-3}$) were about 99%, 98%, 97% and 65%, respectively | A two-phase BTF showed better performance and supported a richer as well as more uniform microbial community in the biofilter bed than a single-phase BTF | [86] |

| VOCs Mixture | Bioreactor Dimensions ($d_{in}$ x H; V) | Packing Material | Inoculum | Process Parameters | Outcomes | Reference |
|---|---|---|---|---|---|---|
| | | Pelletized diatomaceous earth (Celite) | Activated sludge from WWTP as reported in [50] | EBRT: 120 s; 3:1 and 5:1 ratios of methanol to n-hexane; maximum elimination capacity for n-hexane was 11.2 g m$^{-3}$ h$^{-1}$ for its inlet loading of 13.2 g m$^{-3}$ h$^{-1}$ when lower ratio of methanol was tested | Removal of methanol and hexane was favored in acidic conditions and offered high removal efficiencies; degradation of methanol was not affected by n-hexane | [87] |
| Methanol, n-hexane | 0.076 × 1.3; 2.72 | Pelletized diatomaceous earth media support media (Celite) | From previously working BTF From previously working BTF | EBRT: 120 s Acclimation with intermittent loading of methanol Inlet loading of hexane from 21.5 to 47.7 g m$^{-3}$ h$^{-1}$ | Acidic conditions (pH = 4) with a dominant fungi consortium offered higher performance than the process realized at pH = 7; a change in the inlet loading of VOCs may affect the microbial community, especially for high VOC loads | [88] |
| | | | | EBRT: 120 s Inlet loading: hexane 13.2, methanol 37.7 g m$^{-3}$ h$^{-1}$ | Methanol introduction ameliorated the removal of hexane from air, a high reaction rate constant for hexane biofiltration was obtained for the alternate strategy of treatment air polluted with either hexane or its mixture with methanol; the reintroduction of methanol increased the removal of hexane, due to an increase in hexane bioavailability | [89] |
| n-Hexane, benzene, methanol | 0.076 × 1.3; 2.72 | Pelletized diatomaceous earth support media | From previously working BTF, pH = 4 favors fungal growth | EBRT: 120 s Total inlet loading of VOCs between 96.4 and 117.7 g m$^{-3}$ h$^{-1}$; | Results showed that the top section of BTF was the most active part of a biofilter where competition among VOC degraders took place; VOCs concentrations played a role in the recovery of biofilter performance after shut-down periods, as well as affected the removal efficiency of resistant hydrophobic VOCs, such as hexane | [90] |
| Benzene, toluene | Pilot plant; 2.76 × 1; 6000 | Blue mussel shells | Natural from blue mussel shells and WWTP effluent used as a trickling liquid | Gas flow: 0.9 m$^3$ h$^{-1}$; inlet concentrations of benzene and toluene were 0.4–56 and 1.6–22.8 mg m$^{-3}$, respectively | The two-stage process combining a water scrubber and BTF resulted in a high purification degree of air from WWTP | [91] |

**Table 3.** *Cont.*

| VOCs Mixture | Bioreactor Dimensions ($d_{in}$ x H; V) | Packing Material | Inoculum | Process Parameters | Outcomes | Reference |
|---|---|---|---|---|---|---|
| Styrene, acetone | $0.15 \times 1.5$; 17.6 | Polypropylene pall rings | Mixed culture from previously working biofilter | EBRT: 53 s | Investigated system combined a BTF with a downstream conventional biofilter; acetone accumulated in the trickling liquid and hindered the biodegradation; recovery of the BTF performance was possible when acetone in liquid was degraded; a proposed two-stage biofiltration system is promising for the smoothing variation of concentrations or for the overloading of a biofilter unit | [92] |
| Methanol, toluene, trichloroethylene (TCE) | $0.076 \times 1.3$; 2.72 | Celite | Fungi; from previously working BTF | pH = 4 EBRT: 120 s Inlet loading: TCE ($10$–$40$ g m$^{-3}$ h$^{-1}$), methanol ($22$–$237$ g m$^{-3}$ h$^{-1}$), toluene ($17$–$101$ g m$^{-3}$ h$^{-1}$) | The elevated inlet loading of a primary substrate (i.e. carbon source like methanol) inhibited TCE removal; biofilters with lower ratios of methanol and toluene to TCE resulted in better removal of TCE, as well as better responses to increased TCE loads | [93] |
| Trichloro-ethylene (TCE), methanol | $0.076 \times 1.3$; 2.72 | Celite | Fungi; from previously working BTF | pH = 4 EBRT: 120 s Inlet concentrations: TCE ($28$–$80$ ppmv), methanol ($103$–$711$ ppmv) | Higher concentrations of methanol in the inlet stream allowed for the higher removal of TCE, but only for methanol concentrations below the inhibition level; TCE was more bioavailable for higher methanol concentrations; preferential degradation of hydrophilic over hydrophobic substrate was observed; elimination efficiency of methanol was not affected by variations in TCE concentration | [94] |

**Table 3.** *Cont.*

| VOCs Mixture | Bioreactor Dimensions ($d_{in}$ x H; V) | Packing Material | Inoculum | Process Parameters | Outcomes | Reference |
|---|---|---|---|---|---|---|
| Benzene, toluene, ethylbenzene, xylene | $0.084 \times 0.45; 2$ | Kaldnes rings | Activated sludge from denitrification-nitrification section from Valladolid WWTP | EBRT: 30 min Inlet loading of xylene, toluene and ethylebenze was 1.4 g m$^{-3}$ h$^{-1}$ for each and 1.5 g m$^{-3}$ h$^{-1}$ for benzene | Oxygen-free conditions of biodegradation were applied; REs above 90% were noted for toluene and ethylbenzene (relative ease of biodegradation compared to benzene and xylene); removal of xylene was limited due to mass transfer limitations; benzene removal was poor due to toxic intermediate products of biodegradation which were overcome by the application of UV to trickling liquid; the coupling of BTF with a UV gas pre-treatment didn't increase the removal of BTEX in the system; low similarity of inoculum and developed microbial community composition was detected | [95] |
| Cyclohexane, methyl acetate | $0.11 \times 1; 4.75$ | Volcanic rock and ceramsite | Activated sludge from the secondary sedimentation tank from Beijing Gaobeidian WWTP | EBRT: 88.3 s Inlet concentrations of cyclohexane and methyl acetate are up to 250 and 800 mg m$^{-3}$ | REs of over 90% for both investigated VOCs | [96] |
| Toluene, formaldehyde, benzo-a-pyrene | $0.079 \times 1.05$ | Vermiculite | *Fusarium solani* B1 (fungi) and *Rhodococcus erythropolis* DSM 43066 | EBRT: 31 s Inlet loadings: benzo-a-pyrene (373 g m$^{-3}$ h$^{-1}$), toluene (33.5 g m$^{-3}$ h$^{-1}$), formaldehyde (34.8 g m$^{-3}$ h$^{-1}$) | Elimination capacities for benzo-a-pyrene, toluene, and formaldehyde reached 215, 31, and 22.5 g m$^{-3}$ h$^{-1}$, respectively, for conditions indicated in reference conditions given to the left; the biofilter was divided into three sections, and the first stage was preferentially colonized mainly by fungi, while the other two were colonized with bacteria | [39] |
| Chloroform, dichlorobromo-methane, ethanol | $0.076 \times 1.3; 2.72$ | Celite | From previously working BTF | EBRT: 5 min Inlet concentrations of trihalomethanes (total) 14 ppmv, inlet concentration of ethanol 25–200 ppm | The application of a co-metabolite (ethanol) resulted in a similar increase in the elimination of trihalomethanes as the use of a surfactant; natural surfactant was found to be more efficient in aiding the BTF performance than the synthetic one | [97] |

**Table 3.** *Cont.*

| VOCs Mixture | Bioreactor Dimensions ($d_{in}$ x H; V) | Packing Material | Inoculum | Process Parameters | Outcomes | Reference |
|---|---|---|---|---|---|---|
| Toluene, ethyl benzene, p-xylene, m-xylene, o-xylene | $0.1 \times 0.38$; 3 | Ceramic particles | Mixed acclimated microbial consortia from petroleum polluted soil | EBRT: 60 s Inlet concentrations of each VOC from 50 to 160 mg m$^{-3}$ | Continuous and discontinuous BTF feeding strategies were compared; continuous feeding resulted in the higher removal of VOCs than the discontinuous mode, while switching from discontinuous to continuous resulted in a great increase in process efficiency | [98] |
| n-Butanol, cyclohexane | $0.08 \times 0.68$; 2.5 | Peat, perlite, ceramic Raschig rings | No additional inoculation (only microbes naturally occurring in peat material) | EBRT: 46–60 s Inlet loading of n-butanol (19.8–99 g m$^{-3}$ h$^{-1}$) and cyclohexane (45–180 g m$^{-3}$ h$^{-1}$) | Presence of butanol increased the removal efficiency of cyclohexane, starvation episodes slightly decreased biofiltration performance, and the longer the starvation period, the longer time that was needed for the recovery of removal efficiency | [99] |
| Ethanol, hexane | $0.08 \times 0.68$; 2.5 | Peat, perlite, ceramic Raschig rings | No additional inoculation (only microbes naturally occurring in peat material) | EBRT: 60 s Inlet loading of ethanol (18.45–38.5 g m$^{-3}$ h$^{-1}$) and hexane (25–140 g m$^{-3}$ h$^{-1}$) | The presence of ethanol increased the removal efficiency of hexane; ethanol starvation caused only a slight decrease in removal efficiency of hexane; a lower than previously reported volume by volume ratio of ethanol ensured the efficient biofiltration of hexane | [100] |
| Cyclohexane, ethanol | $0.08 \times 0.68$; 2.5 | Polyurethane foam | *Candida albicans*, *Candida subhashii* | EBRT: 60 s Inlet loadings of VOCs in the range from 36 to 90 g m$^{-3}$ h$^{-1}$ | Feeding a BTF with a mixture of ethanol and cyclohexane resulted in the higher removal of cyclohexane compared to when ethanol was added to the gas stream polluted with cyclohexane only; negligible concentrations of VOCs were detected in the liquid phase in steady-state conditions | [101] |

**Table 3.** *Cont.*

| VOCs Mixture | Bioreactor Dimensions (d$_{in}$ x H; V) | Packing Material | Inoculum | Process Parameters | Outcomes | Reference |
|---|---|---|---|---|---|---|
| n-Hexane, dichloromethane | No information | Bamboo charcoal based polyurethane foam | Activated sludge from the pharmaceutical factory WWTP | EBRT: 20–150 s Inlet concentrations of n-hexane (100 mg m$^{-3}$) and dichloromethane (150 mg m$^{-3}$) | The applied packing material offered large and uniform pores, thus improving the mass transfer from gas to liquid phase; further, the modification of polyurethane foam with charcoal increased its porosity, resulting in greater surface area available for biofilm development; the packing offered higher resistance to transient conditions | [42] |
| m-Xylene, toluene | 0.0885 × 0.7; 2.4 | Diatomaceous earth pellets | Acclimated activated sludge | EBRT: 25–60 s Inlet concentrations of m-xylene (250–1500 mg m$^{-3}$) and toluene (250 mg m$^{-3}$) | The mechanisms of synergistic removal of m-xylene in the presence of toluene were discussed in terms of toxicity effects, co-metabolism, and suitable concentration proportions (1:2 for toluene to m-xylene) | [49] |
| Toluene, hexane, α-pinene, trichloroethylene (TCE) | 0.1 × 0.34; | Polyurethane foam | *Candida subhashii* | EBRT: 30 s Inlet concentrations of VOCs in the range from 200 to 450 mg m$^{-3}$; removal efficiency in the range 20–45% | A biodegradation pattern for investigated VOCs was found: toluene > n-hexane > α-pinene > TCE; this sequence was discussed in terms of Hansen solubility parameters; biotrickling filter configuration proved to be more efficient in biodegradation of VOCs than the conventional biofilter | [102] |

## 4. Modeling of Biofiltration and Biotrickling Filtration Processes

The use of mathematical models can help to explore the phenomena occurring during various processes. The modeling of a bioprocess is a representation of the biological, chemical, and physical processes occurring in the bioreactor, and aims at the selection and optimization of several process parameters that can affect the process performance. Mathematical modeling helps in understanding the process, as well as aids in the reactor design and scaling up [69,103]. In fact, modeling is a means of translating concepts and observations into mathematical equations [104–106]. Process modeling should indicate or help to identify the expected course of the process as a result of changes introduced among the system variables.

The approaches towards the process modeling of conventional biofiltration and biotrickling filtration are different; however, they share several common features. According to the development of biological methods for air treatment, the modeling of biofiltration in conventional biofilters was initiated earlier than for biotrickling filters. The main differences in the modeling approaches result from differences in the process, reactor design, and operation. Conventional biofilters usually use organic media, and no regular flow of a liquid through a packing occurs. Additionally, a treated gas is pre-humidified prior to entering the biofilter. In biotrickling filters, inert materials are used as packing elements and

a continuous or intermittent flow of liquid occurs through the packing. However, in practice, there are a mix of above-mentioned features in both biofilters and biotrickling filters. For example, biofilters require additional watering of the packing, and newly developed packing media are being applied [56,107]. Thus, various similar phenomena take place in both the BF and BTF, which allows for similar descriptions when using mathematical modeling [108].

Both the BF and BTF involve a complex system of various aspects of chemical, physical, and biological processes. Thus, plenty of elements must be taken into consideration when proposing a mathematical model of these processes, including mass balances for treated compounds, oxygen, and products of biodegradation ($CO_2$, biomass, other intermediate products) with respect to different phases within a bioreactor i.e., the gas phase, the solid phase with biofilm, and the liquid phase [104]. Additionally, in order to set up some simplifications (model assumptions) and develop a reliable model, various problems and phenomena must be taken into account. Among others, these include oxygen limitation, estimation of available interfacial area, adsorption on the solid phase, biodegradation aspects with kinetic or mass transfer limitations, uniformity wetting of both packing and biofilms, and interactions between target compounds when VOCs mixtures are treated.

The modeling of conventional biofiltration is well described in the literature. However, the modeling of biotrickling filtration is distinctly different than that of conventional biofiltration. One of the available reviews of mathematical models for biotrickling filtration was published by Deshusses and Shareefdeen [104]. The authors revised the models for both conventional biofiltration and biotrickling filtration, and stressed the differences and complications regarding the modeling of biotrickling filtration. The authors state that, besides the fact that BTF includes one more phase to be modeled (i.e., liquid phase) than BF, some aspects of modeling the BTF may be even easier than for the BF. Shareefdeen and Singh [109] point out that the main differences that make the modeling of BTFs more complicated include the existence of the liquid phase, the increased complexity of the process, and the various possible mass transfer pathways (direct transfer of the target pollutant to the biofilm or indirect to the biofilm via liquid phase). On the other hand, due to the utilization of inert materials, such as packing elements, it is possible to determine the active surface area of the packing more easily and precisely. It is also possible to determine other parameters, such as the concentration in the liquid phase or in the biofilm, which are hardly measured for biological packing in conventional biofilters. Another advantage is related to the possibility of more in-depth biofilm analysis, due to its macroscopic character (biofilm is usually clearly visible).

However, this paper deals with biotrickling filtration. Due to the historical perspective, similarities, and initial bases for the development of a mathematical description of the biofiltration processes, an overview of conventional biofiltration modeling is also provided.

### 4.1. Modeling of Conventional Biofiltration

In accordance to the historical development and innovation in biological treatment of polluted gases, modeling of such processes started from the modeling of conventional biofiltration. The development of biofiltration models was directly connected to results from the progress of knowledge and understanding of complex phenomena occurring during the biofiltration process. The first mathematical model of biofiltration was proposed by Ottengraf and van den Oever [110]. This model served as the basis for many models of biofiltration, and the following assumptions are undertaken in this model:

- target compounds (i.e., compounds removed from air in biofiltration system) are transported within the biofilm layer via diffusion, described by an effective diffusion coefficient;
- the biofilm thickness is much smaller that the diameter of a bed particle;
- the biodegradation of target compounds in the biofilter can be described by the Michaelis–Menten equation;
- the flow of the gas phase is a plug flow;

- mass transfer resistance at the gas side in the gas-biofilm interface is negligibly small.

Deshusses and Hamer [111] provided summary formulae for the Ottengraf and van den Oever model while considering three operational conditions that depend on the biodegradation kinetics. These operating situations are valid for the bottom of an expanded clay sphere-packed biofilter with a downward flow of gas phase:

$$\frac{C_G}{C_{G,in}} = \exp\left(\frac{hK_1}{m_i u_G}\right) \tag{5}$$

$$\frac{C_G}{C_{G,in}} = 1 - \left(\frac{hK_0}{u_G C_{G,in}}\right) \tag{6}$$

$$\frac{C_G}{C_{G,in}} = \left[1 - \frac{h}{u_G} \cdot \sqrt{\frac{K_0 D_e a}{2 m_i C_{G,in} \delta}}\right]^2 \tag{7}$$

where $a$ (m$^2$ m$^{-3}$) is an interfacial area per unit volume, $C_G$ (mg m$^{-3}$) is a component concentration in the gas phase, $D_e$ (m$^2$ s$^{-1}$) is an effective diffusion coefficient, h (m) is the height of the bed, *in* refers to the biofilter inlet, $K_0$ (g m$^{-3}$ s$^{-1}$) and $K_1$ (s$^{-1}$) are zero- or first-order reaction rate constants, respectively, $m_i$ is a distribution coefficient of a component *i*, and $u_G$ (m s$^{-1}$) is the superficial gas velocity.

Shareefdeen and Baltzis [112] stated that the assumptions from steady-state conditions cannot be directly applied to transient conditions, which is true both for conventional as well as biotrickling filters.

Based on the available literature, in 2002 Świsłowski [113] concluded that the biofiltration process, together with possibilities to predict the process performance, was valid for steady-state conditions only. Świsłowski postulated that much less data were available on the unsteady-state conditions, e.g., when the biofilter is undergoing process start-up and stabilization. An empirical relationship was proposed for the determination of the liquid–solid mass transfer of target compounds in the liquid phase of the biofilm.

Dupasquier et al. [114] investigated the effects of inlet gas mixture composition and possible co-metabolic mechanisms for the removal of methyl tert-butyl ether and pentane, and proposed a model describing the substrate / co-substrate biodegradation of VOC mixtures in a biofilter.

Deshuses and Shareefdeen [104] proposed a general division of biofiltration models into steady-state models and transient models. They examined the Ottengraf and van den Oever model with further modifications within the group of steady-state models. These models assume 1st or 0th order biodegradation that can be described by Michaelis–Menten kinetics. Additionally, two possible cases can be identified, depending on factors such as microbial consortia in the biofilter or treated compounds. The other group of models is the transient models. Shareefdeen and Baltzis [112] proposed a model taking into account the adsorption phenomena. They concluded that models describing steady-state conditions are not applicable to transient conditions, e.g., during the biofilter start-up or variations in process conditions, which happen frequently in practical applications. This is because of two problems: the adsorption and absorption of VOCs at the packing and in the liquid phase, as well as the problem of non-uniform biomass distributions, especially during the development of a biofilm during the start-up period.

Hodge and Devinny [115] developed a model describing basic transport and biological processes for a biofilter treating ethanol vapors. The model included a transfer of ethanol between air and solid/water phases, the biodegradation of ethanol, the resulting $CO_2$ production, and the pH changes due to the accumulation of $CO_2$. These pH changes are related to acidification, and are caused by the uptake of $CO_2$ and its partial oxidation to acetic acid. A model incorporated axial dispersion and excluded the plug flow assumption, while no large-scale turbulence within a biofilter, homogeneity of filter material composition, uniform biomass distribution, first-order biodegradation kinetics, or stoichiometric

production of $CO_2$ from ethanol and oxygen were assumed. The following relationship for ethanol concentration in air phase was proposed:

$$C_G = C_{G0} exp\left(\frac{-b_1 k_m x}{Z}\right) \tag{8}$$

where $C_{G0}$ (mg cm$^{-3}$) is the initial ethanol (i.e., substrate) concentration in the gas phase, $b_1$ (h$^{-1}$) is a first-order biodegradation rate constant, $k_m$ is the ratio of a substrate mass in solids/water phase to its mass in the gas phase, $x$ (cm) is a coordinate position in the biofilter, and $Z$ (cm h$^{-1}$) is the axial interstitial velocity in the gas phase.

In 1995, a new approach to biofilter modeling was proposed by Deshusses et al. [116]. This approach concentrated on dynamic conditions during the biofiltration process, and the model considered a biofilter as a bioreactor comprised of several finite sections, for each of which transient mass balances can be determined. Competitive inhibition kinetics were incorporated in the model and described by the following expression:

$$R_{s\ j,n,w} = \frac{V_{mj} S_{j,n,w}}{K_{mj}\left(1 + \frac{I_{n,w}}{K_i}\right) + S_{j,n,w}} \tag{9}$$

where $R_{sj}$ (kg m$^{-3}$ s$^{-1}$) is a degradation rate constant, $V_m$ (kg m$^{-3}$ s$^{-1}$) is a maximum degradation rate, $S_j$ (kg m$^{-3}$) is the concentration of competitive substrate (i.e., biodegradation inhibitor), $K_{mj}$ (kg m$^{-3}$) is the Michaelis–Menten constant of component $j$, $K_i$ (kg m$^{-3}$) is the inhibition constant of $I$ on the removal rate of $j$; and subscripts $n$ and $w$ denote biofilm/sorption volume subdivisions and biofilter layer subdivisions, respectively.

In 1997, Shareefdeen proposed a general transient biofilter model for single and multiple VOC removal from air, which included a so-called interactive kinetics assumption, multi-component adsorption phenomena, axial dispersion effects, and oxygen limitation. Amanullah et al. [117] proposed a model taking into account convection and dispersion phenomena in the gas phase, partial coverage of the solid support by the biofilm, interphase mass transfer between the gas phase and the aqueous biofilm with an equilibrium partition at the interface before diffusion, direct adsorption to the available uncovered solid adsorbent media, transfer between the biofilm and the solid support, and the biological reactions in both the biofilm and the adsorbent. The authors pointed out that the model can be successfully applied to model conventional biofiltration for a wider range of process parameters than previously proposed models. Additionally, it was postulated that a considerable role in the elimination of VOCs from the gas phase may be attributed to adsorption of support elements not covered by biofilm. The adsorbed compounds may then undergo biodegradation, or at least such adsorption may help in handling the fluctuating inlet loads.

The problems of biomass overgrowth and biofilter clogging were investigated and modeled by Ozis et al. [118]. This model of biofilm growth was based on the combination of percolation theory and a cellular automaton model.

Further development of biofiltration modeling includes increasingly broader use of software for the modeling and control of industrial-scale biofilters [104,119], and neural networks are being used for the development of models with dynamic conditions.

A model combining dynamic conditions during biofiltration with biomass growth and resulting pressure drops was proposed by Xi and co-workers [120]. Vaiškūnaitė and Zagorskis [121] developed a model for the treatment of VOC mixtures. Recently, the influences of substrate degradation and inhibition mechanisms were investigated and modeled [122].

A brief overview of models proposed for the description and prediction of conventional biofiltration processes, including the most important assumptions and simplifications, is given in Table 4.

**Table 4.** Overview of selected mathematical models for conventional biofiltration.

| Target Compound(s) | Process Parameters | Model Description | Reference |
|---|---|---|---|
| Mixture: toluene, butylacetate, ethylacetate, butanol ($C_{in}$ for each component ranged from 0.050 to 5.606 g m$^{-3}$) | 5-stage Quick fit glass columns ($d_{in}$ = 0.1524 m, H = 0.6 m) Peat compost inoculated with toluene degrading organism $u_G$: 0.84 and 14.8 cm s$^{-1}$ | Steady-state theoretical model describing the elimination of the carbon sources in the filter bed; Monod equation is adapted for the microkinetics description, and model supports identification of either reaction or diffusion limitation | [110] |
| Methanol ($C_{in}$ about 6.4 g m$^{-3}$) | Peat-perlite packed biofilter ($d_{in}$ = 0.05 m, H = 0.6 m) $u_G$ = 6.4–12.7 m h$^{-1}$ | The model considers the consumption of two substrates i.e., methanol and oxygen; the diffusion and reaction of these two substrates in the biofilm at quasi-steady state conditions are considered with Andrews and Monod-type dependences on the concentrations of methanol and oxygen, respectively; the model assumes gas plug flow and uniform biofilm density along the biofilter; the model showed that the biofiltration process is mass-transfer limited by oxygen and kinetic-limited by methanol | [123] |
| Toluene (IL range: 4.8–26.8 g m$^{-3}$ h$^{-1}$; $C_{in}$: 0.6–2.8 g m$^{-3}$ | Plexiglass column ($d_{in}$ = 0.102 m, H = 0.686 m), sterilized peat and perlite inoculated with toluene-degrading consortium | The model approaches the description of biofiltration at transient conditions; the model assumes that the biolayer may not be uniform at the surface of packing elements, and VOCs can be reversibly adsorbed on the packing elements not covered with biofilm; the biodegradation rate depends on the concentrations of both VOCs and oxygen, and can be determined based on batch suspended culture experiments; the air stream flow is a plug flow | [112] |
| Benzene, toluene | Plexiglass column ($d_{in}$ = 0.1 m, H: 0.51 m for benzene and 0.69 for toluene); Contaminated soil-based microbial consortium inoculated on peat-perlite particles | The model attempts to predict biofilter capabilities and aid during the biofilter design stage; the model describes and predicts with good accuracy steady-state biofiltration; results of the modeling and experiments revealed that benzene is harder to remove from air than toluene, and the biofiltration of hydrophobic compounds is less oxygen-limited than in the case of hydrophilic VOCs | [124] |

| Target Compound(s) | Process Parameters | Model Description | Reference |
|---|---|---|---|
| Ethanol | Polyvinyl chloride biofilter column ($d_{in}$ = 0.076 m; H = 0.9 m); three packing media were compared: granular activated carbon, sintered diatomaceous earth, and compost | Development of a model by Shareefdeen and Baltzis using exclusion of a plug-flow assumption and the incorporation of axial dispersion in transient conditions; the model provided foundations for further biofilter developments, including the most important feature affecting the biofiltration performance i.e., packing media allowing for enhanced biodegradation, with the lowest fraction of area available for microbial growth, and which allows for high adsorption capacity for the treated compound | [115] |
| Methyl ethyl ketone, methyl isobutyl ketone and their mixture | Plexiglass column ($d_{in}$ = 0.08 m, H = 1 m); Bioton packing (mixture of compost and polystyrene spheres) | The proposed dynamic, diffusion reaction model describes both the transient and steady-state biofiltration performance, including kinetics with substrate competition | [116,125] |
| Benzene, toluene | Model was verified using data from [124] | A general transient biofiltration model incorporating mixing, oxygen limitation, and adsorption phenomena; the biofilm is assumed to not necessarily be uniformly formed over the bed particles; planar geometry of biolayer, reversible VOC adsorption and no biomass accumulation are assumed; | [126] |
| Binary gas mixtures: benzene-toluene or ethanol-butanol | 3-sectional glass biofiltration column ($d_{in}$ = 0.152 m, H = 0,915 m), packed with a mixture of peat moss and horticultural grade perlite | Steady-state model for the biofiltration of binary VOC mixtures; the model incorporates potential kinetic interactions between the air pollutants (VOCs i.e., substrates), as well as the effects of oxygen limitation on biodegradation and biomass diversification in the biofilter packed bed; results suggest that the modeling of VOC mixture biofiltration is not straightforward when compared to modeling single compound biofiltration | [47] |
| Model validation on selected literature data | $A_s$ = 0.457–1.9 $Pe$ = 1–1000 $m_{2,I}$ = 0.002–20 | A transient model in which dispersion phenomena, convection, adsorption, biomass diversification, oxygen limitation, diffusion, and biodegradation in both the biofilm and a solid support are considered. | [117] |

| Target Compound(s) | Process Parameters | Model Description | Reference |
|---|---|---|---|
| Methanol, $\alpha$-pinene | Plexiglass, 4-segment biofilter ($d_{in}$ = 0.28 m, $H_{total}$ = 1.2 m); compost and wood chips as packing materials | A model approaching the description of biofiltration of a mixture of hydrophobic and hydrophilic VOCs; the steady-state model, including Monod kinetics with inhibition, showed that the prediction of biofiltration performance for $\alpha$-pienene as a representative of hydrophobic VOCs can be achieved by using an air/biofilm partition coefficient instead of a typical air/water partition coefficient, due to lipophilic characteristics of the biofilm | [127] |
| Hydrogen sulfide | Pilot-scale biofilter made of a fiberglass cylinder: $d_{in}$ = 0.608 m; H = 1.2 m | The Biofilter$^{TM}$ two-phase model was used; homogeneity of biofilm was assumed; the model has been incorporated into a software | [119] |
| Butanol, methyl ethyl ketone | Literature data were used for model validation | The model uses a linear driving force approach, assumptions include axially dispersed plug flow for the gas phase, pollutant adsorption phenomena, and no oxygen limitation, as well as first-order kinetics of biodegradation | [128] |
| Hydrogen sulfide | Industrial-scale biofilter (volume of packing: 16.12 m$^3$; H = 1.524 m); Biosorbens (inorganic media) packing material | Use of biofilter models in real full-scale applications using a dedicated software | [104] |
| Ethanol | Acrylic plastic biofilter column ($d_{in}$ = 0.07 m, H = 0.25 m), sand and lava rock inoculated with raw sludge acclimated to ethanol degradation | The model combines a percolation theory and a cellular automaton model for the description of biofilm growth; it was found that, during ethanol biofiltration, oxygen concentration was a limiting factor in the upper parts of the biofilter, while ethanol concentration was limiting in the bottom regions (the supply of treated air from top of the biofilter); the model allows for predicting decreases in biofiltration performance with respect to increasing pressure drops across the packing | [118] |
| Styrene | Perlite inoculated with a mixed microbial culture from petrochemical refinery sludge | A back propagation neural network algorithm was used to model and predict biofiltration performance based on the inlet styrene gas-phase concentration and gas flow rates | [129] |

**Table 4.** *Cont.*

| Target Compound(s) | Process Parameters | Model Description | Reference |
|---|---|---|---|
| Methyl propyl ketone, toluene, p-xylene, n-butyl acetate | Model was validated on literature experimental data | A dynamic model for predicting biofiltration performance; assumptions include gas phase plug flow, utilization of Henry's law to describe gas-biofilm interface equilibrium, Fick's law to describe diffusion, planar biofilm geometry, and adsorption phenomena | [130] |
| Toluene | A biofilter column: $d_{in}$ = 0.12 m; H = 0.4 m; packing material: wood chips and propylene spheres | Biofiltration model incorporating biofilter performance as a function of biomass growth and pressure drop for varied toluene loadings; toluene biodegradation follows Monod kinetics, biofilm is treated as a flat surface; in contrast to previously described models, this model employs microbial growth kinetics, including inert biomass formation, and the model has been verified for various flow rates and concentrations, which mimic the real variations in gas effluents | [120] |
| Butyl acetate, butanol, xylene | A biofilter with dimensions: 0.5 × 0.48 × 2 m (length: width: height) packed with activated pine bark | The proposed mathematical model aims at predicting the biofiltration performance of VOC mixtures depending on the mixture composition and ratio of investigated VOCs; surface response methodology and finite difference approximation were employed to deliver model solutions; relatively simple equations were proposed to determine the elimination capacities of each studied VOC | [121] |
| Methanol, α-pinene | Model validation is based on the results obtained by Mohseni and Allen [127] | The model gives approximate analytical expression of the concentration profiles of studied VOCs in air/biofilm phases using the Adomian decomposition method; non-linear differential equations have been solved analytically in this biofiltration model | [131] |
| Methanol, α-pinene | Model validated on experimental data from literature | An approach of a biofilter model based on expressing the particular solution of governing equations using an integral containing Green's function; the proposed model is a development of a model proposed by Mohseni and Allen [127] | [132] |

**Table 4.** *Cont.*

| Target Compound(s) | Process Parameters | Model Description | Reference |
|---|---|---|---|
| Toluene | Polycarbonate biofilter column ($d_{in}$ = 0.1 m, H = 0.63 m) packed with sterilized compost and wood chips, inoculated with *Nocardia* sp. | A hypothesis was formulated that a biofilter can progress from a single steady-state to a multi steady-state, driven by the growth of biofilm thickness and biofilm die-off during biofiltration; toluene removal was decreased with biofiltration time for increasing inlet toluene concentrations, which was proven to be related with the shift from a diffusion-limited biofilm high-activity state to a low-activity state; this can be a result of substrate degradation and consequent inhibition following Haldane kinetics and pollutant diffusion within the biofilm | [122] |

$u_G$—superficial gas velocity; $A_s$—biofilm surface area per unit volume of particle; Pe—Peclet number (ratio of convective to diffusive mass transfer rates; product of Reynolds and Schmidt numbers); $m_{2,i}$—distribution coefficient of substance *i* in an air/solid media system.

### 4.2. Modeling of Biotrickling Filtration

Due to the complexity of various physical, chemical and biological processes that constitute biotrickling filtration, its mathematical modeling is challenging; however, such modeling may reveal important information about the biotrickling filtration mechanisms and process optimization strategies [103].

The beginning of biotrickling filtration modeling is related to works by Hartmans, Dick, and co-workers regarding the steady-state removal of dichlorobenzene [133,134]. Mpanias and Baltzis proposed a model taking into account kinetic and oxygen limitations during the biodegradation of VOCs [135]. Okkerse and co-workers proposed a dynamic model incorporating the possible acidification of a trickling liquid, as well as biomass accumulation and distribution along the biofilter [136]. Dynamic models, including the mass transfer limitations of hydrophobic VOCs in aqueous systems and non-uniform biofilm wetting, were then developed [137,138]. Further, models, including the substrate competition and the cross-inhibition of substrates during biotrickling filtration were developed [139,140]. The effects of possible absorption of VOCs by the liquid phase were included in the model proposed by Hernandez et al. [141]. The modeling of biotrickling filtration for mixtures of hydrophilic volatile organic compounds, including the effects of intermittent trickling patterns, were proposed by San-Valero and co-workers [103]. Recently, Kalantar et al. [142] proposed a dynamic model for the biotrickling filtration of mixed hydrophilic and hydrophobic VOCs. Brief overviews of state-of-the-art biotrickling filtration models for the time period before 2018 were provided by Sharvelle et al. [143], Ahmed et al. [144], San-Valero et al. [103], and Gąszczak et al. [145]. A few review descriptions on biotrickling filtration models are also available in other papers referred to in this work.

The modeling of the biotrickling filtration process, which is similar to conventional biofiltration, requires knowledge on the courses of single sub-processes, such as mass transfer of the target compound, oxygen limitation from the gas to liquid/biofilm phases, biodegradation kinetics, and hydrodynamic behavior of the bioreactor. San-Valero et al. [146] determined the mass transfer coefficients for isopropanol and oxygen for biotrickling filters packed with random or structured media, and proposed the use of a global mass transfer coefficient to describe the mass transfer phenomena, which included a simple methodology to aid in the mathematical modeling of the biotrickling filtration of hydrophilic VOCs.

*4.3. Problems and Challenges for Modeling of Biotrickling Filtration*

The following problems and challenges can be identified when modeling biotrickling filters:

- non-uniform wetting of the packing and biofilm
- oxygen limitation
- biokinetic or mass transfer and diffusion limitations, depending on the substrate type
- stripping of pollutants at the top of the biotrickling filter (in the vicinity of a trickling liquid sprayer)
- existence of active and inactive biomass zones
- determination or estimation of the gas-biofilm/liquid interface area
- the need to include gas–liquid / gas–liquid–biofilm mass transfer as well as adsorption of VOCs in the solid phases; however, problems with adsorption may be only relevant for rare package materials, as most materials are made of plastics with low adsorption capacity
- interactions between target compounds.

An important aspect of biotrickling filtration modeling is related to the description of biodegradation kinetics. Kinetics is one of the most important parameters affecting the process rate and performance, as well as expected model fitting to experimental data. In order to model the performance of a biotrickling filter, or any other biofilter type, it is necessary to assume a rate equation describing the consumption rate of the substrate by microorganisms [147]. This is an important step of model development. Mpanias and Baltzis proposed a model suitable for any VOC for which the biodegradation kinetics are known [135]. The following most important assumptions were undertaken in this model and are frequently valid for many other models: steady-state conditions are assumed; VOC and oxygen are transferred from the gas to liquid phase, and then from liquid to biofilm; biodegradation of the VOC takes place in the biofilm layer; the biodegradation rate is limited only by the concentrations of VOC and oxygen; the biodegradation takes place in the biolayer (bioactive layer) of the biofilm; anaerobic degradation of the VOC is not considered; concentrations of VOC and oxygen in gas/liquid phases follow Henry's law; and the concentrations of VOC and oxygen in the biofilm are the same as in the liquid phase.

Assumptions undertaken in models for conventional biofilters and biotrickling filters are similar to some extent, e.g., kinetics and biodegradation description. The clearest difference in the approach to modeling of these processes is due to mass transfer from gas to biofilm phases. Usually, in biotrickling filters, it is assumed that the VOC is firstly transferred to the liquid phase, and then to the biofilm, which may result in, peak concentrations of the VOCs that can biodegrade, depending on the trickling mode (continuous or intermittent) [103].

The approach and means of BTF modeling have been developing, as indicated by selected milestones reported in Figure 1, as seen in the perspective of the overview of process modeling, which is given in detail in Table 5. As with conventional biofiltration, steady-state models were initially proposed, followed by the dynamic models, which were more suitable for the transient process conditions. A common feature during the development of each mathematical model of biotrickling filtration, especially for dynamic models, included the definition of mass balances for gas, biofilm, and possible liquid phases. These were the initial steps during the formulation of phenomenological models [69]. Model development also requires accepting the description of diffusion phenomena and biodegradation kinetics, as well as defining limitations (e.g., oxygen limitation). All this data are reflected in various model parameters, that either need to be experimentally determined for a specific system, or may be calculated or accepted based on the available literature. It must be stressed that several parameters may be compound-specific or biofilm/microbial-type specific, and thus require individual revision or calculation. Another important step of model development includes model calibration and validation by using either new or available experimental data. Upon the development of new models and the inclusion of more assumptions and

limitations, the models became more and more complex, and seldom resulted in a single equation allowing for one result, such as the determination of VOC concentrations in treated gas leaving the biofilter. Due to the fact that the majority of models revised in Table 5 are phenomenological models, together with their complexity and usually individual character, specific model equations regarding computational values, such as mass balances, diffusion coefficients, or formulae for model functions were intentionally omitted in this review. In the case of interest in the specific model, the reader is kindly asked to refer to the original source of the literature reference.

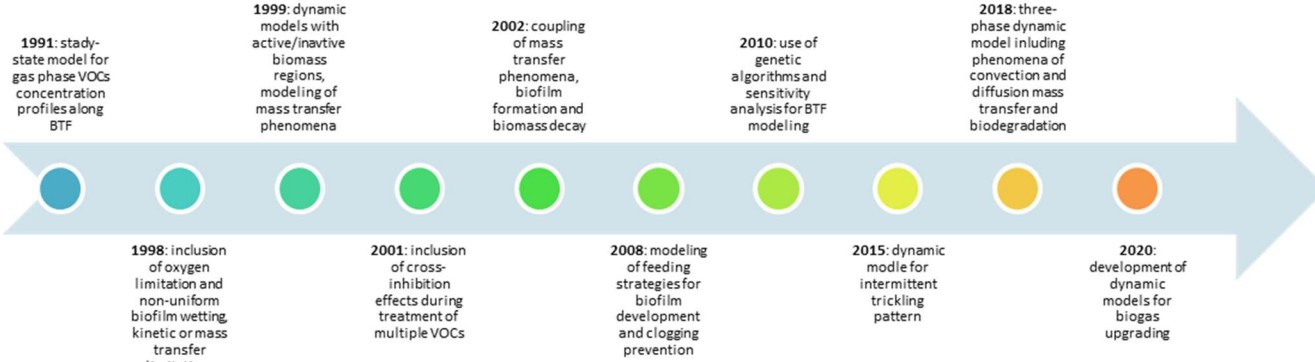

**Figure 1.** Milestones in the development of mathematical models for biotrickling filtration of air polluted with VOCs.

The progress in BTF modeling has mainly included new assumptions to make the models more closely replicate practical/industrial applications and problems. However, among plenty of models, similar approaches are common, including those regarding the gas phase flow pattern, distinguishing the contacting phases and interfaces, and biodegradation kinetics descriptions. The last aspect, i.e., biodegradation kinetics, seems to be of special importance, due to the biological character of biotrickling filtration; thus, recent models include multiple substrate approaches or induction/inhibition effects [148,149]. Currently, the development of biotrickling filtration and process modeling is bieng observed in the field of biogas upgrading [150–152]. This reflects a close and reasonable relationship between scientific studies and the current need for the management and utilization of natural and renewable resources, for which biotrickling filtration can be of special importance.

**Table 5.** Overview of selected mathematical models for biotrickling filtration.

| Target Compound (s) | Biofilter Parameters | Model Description | Reference |
|---|---|---|---|
| Dichlorobenzene | Vertical Perspex pipe ($d_{in}$ = 0.29 m; H = 1 m) packed with polypropylene elements (Filterpack) inoculated with adopted dichloromethane degrading microorganisms | Simulation of the biotrickling filter as a series of perfectly stirred interconnected reactors; computer software was used for calculating dichloromethane conversion using immobilized biocatalysts; the modeling gave surprisingly good results when compared to experimental data, however no mass transfer resistance between liquid and biofilm was included | [134] |

**Table 5.** *Cont.*

| Target Compound (s) | Biofilter Parameters | Model Description | Reference |
|---|---|---|---|
| Dichloromethane | Ceramic saddles | Steady-state model with the assumption of zeroth order kinetics; negligible resistance for the mass transfer of VOCs from gas to liquid phase; the model allowed for determining concentration profiles along co- and counter-current biofilter | [133,153] |
| Theoretical model | No experimental verification; modeling study | The model incorporates simplified equations describing the gas flow and biodegradation; the model indicated two aspects of biotrickling filtration: the inclusion of both stripping in counter-current operation, and the influence of absorption/biodegradation in the liquid phase | [154] |
| Toluene | Stainless-steel biotrickling filter packed with Celite (pelletized diatomaceous earth) inoculated with activated sludge | A modification of a conventional model for synthetic-media biotrickling filters using one substrate, uniform biomass, and one type of microbial population; uniform pollutant concentrations in the gas phase are assumed; only gaseous and biofilm phases are considered | [155] |
| m-Chlorobenzene | Glass column ($d_{in}$ = 0.152 m; H = 0.74 m) packed with Intalox ceramic saddles inoculated with a stable microbial consortium | The model incorporates oxygen limitation, non-uniform wetting of biofilm, and step-wise mass transfer of pollutants from the gas to liquid to biofilm phase; reactions occur only in the effective biolayer; no anaerobic degradation of VOCs is assumed; constant density of the biofilm is assumed | [135] |
| Pentane, izobutene | A biotrickling filter (din = 0.0508 m; H = 0.5 m) packed with a structured polyethylene packing inoculated with microbial consortium taken from previous studies | Steady-state model assuming a biofilm covered with a liquid phase; a two-step approach of modeling, including dependence on mass transfer and then biodegradation kinetics is proposed; results of the modeling suggest both kinetic and mass transfer limitations during the biotrickling filtration of pentane and izobutene | [156] |
| Acetone, toluene | Stainless-steel bioreactor packed with pelletize diatomaceous earth particles (H = 1.14 m) | Conceptual model incorporating active and inactive regions of biomass as well as the simultaneous growth and detachment of biofilm; modeling results indicated high mass transfer resistance for the liquid phase | [157] |
| Diethyl ether | Glass BTF divided into 7 sections, $d_{in}$ = 0.076 m and total height of packing H = 0.61 m; packing material: Celite (porous ceramic particles); inoculation with enriched microbial culture from activated sludge | The model is a development of [157]; Glass BTF is divided into 7 sections, din = 0.076 m, and total height of packing H = 0.61 m; packing material: Celite (porous ceramic particles); assumes inoculation with enriched microbial culture from activated sludge | [137] |

**Table 5.** *Cont.*

| Target Compound (s) | Biofilter Parameters | Model Description | Reference |
|---|---|---|---|
| Dichloromethane | Glass fiber enforced plastic column, din = 0.396 m and H = 2.7 m; packed with a corrugated sheet 60° PVC cross-flow cooling packing, inoculated with a biomass suspension of *Hyphomicrobium* sp. GJ21 | A dynamic model assuming the development of active (i.e. dichloromethane-degrading microorganisms) as well as inactive biomass; dicholormethane undergoes mass transfer from the gas to liquid phase to biofilm phase; a model for the biodegradation of acidifying VOCs; the model predicts process performance and related gasphase and liquid VOC concentration profiles, as well as the effects of initial VOC concentration increases in a treated gas, like bed clogging | [136] |
| Carbon disulfide | Acrylic tube of din = 0.29 m and height of packing H = 1.2 m; packed with Pastdec 12060 structured PVC packing; inoculation with *Thiobacilii* consortia | Modeling of mass transfer from the gas to liquid to biofilm phase; a single criterion was proposed for identifying the rate-limiting step of biotrickling filtration; a model for describing substrate gas-phase and liquid concentrations and their effects on mass transfer; low substrate concentration in the liquid phase illustrates mass transfer limitations, while higher concentrations suggest kinetic limitation in the biofilm; two-parameter steady-state, one-species, one-directional and heterogeneous model; the model predicts substrate concentration profiles for co- and counter-current modes | [158] |
| Diethyl ether | A four-section BTF packed to the total height H = 0.122 m with Celite (pelletized diatomaceous earth), inoculated with ethyl ether acclimated and enriched aerobic culture, originating from activated sludge | Conceptual model incorporating the reduction of the gas–liquid interface area and the increase in liquid velocity occurring during biomass growth and biofilm development; the paper presents very clear assumptions for the model; a dynamic model for biotrickling filtration was developed, with a focus on nutrient addition and the removal of clogging overgrowing biomass; the model included two effects of backwashing, i.e. biomass thickness reduction and related changes in the reactor specific surface area; the model allows for qualitative description of the biotrickling filtration process | [159] |
| Chlorobenzene, ortho-chlorobenzene | Glass BTF column of $d_{in}$ = 0.152 m and H = 0.79 m packed with Intalox ceramic saddles, inoculated with microbial community from previous studies | The model is a development of [135] and includes cross-inhibition effects during the biofiltration of a mixture of chlorobenzene (with kinetic competition for biodegradation) | [160] |

**Table 5.** *Cont.*

| Target Compound (s) | Biofilter Parameters | Model Description | Reference |
|---|---|---|---|
| Hydrogen sulfide | Pilot scale biofilter—insulated vertical fiberglass cylinder with $d_{in}$ = 0.608 m and H = 1.8 m, packed with lava rock particles, inoculum from activated sludge | Conceptual model incorporating detailed mass transfer phenomena, biofilm formation, and biomass decay; a dynamic model for biofiltration of real waste air streams from wastewater treatment plants | [161] |
| | Differential biotrickling filter filled with a open-pore polyurethane foam cube (side wall of 0.04 m) with already developed biofilm from previous processes | Conceptual model for predicting mass transfer phenomena; the model considers the wetting of packing, as well as gas/liquid/biofilm mass transfer issues; uniform biofilm thickness, non-uniform biofilm wetting, plug flow for gas phase; no reaction in the liquid is assumed; the model proved that the biofiltration of $H_2S$ may be highly limited by the external mass transfer | [138] |
| Mono-chlorobenzene | Perspex tube ($d_{in}$ = 0.05 m; H = 1 m); coal particles on acrylic plastic mesh; inoculation with activated sludge | The model utilizes Monod kinetics for the description of biodegradation; biodegradation is evaluated solely as an aerobic process; the plug flow of gas phase and negligible axial dispersion are assumed, the biofilter is regarded isothermic; the model predicts the concentration profile of the VOC in the gas phase, in the biofilm, and in the trickling liquid retained on solid particles; the model follows first-order kinetics | [162] |
| Dimethyl sulfide and methanol | PMMA biofilter ($d_{in}$ = 0.101 m; H = 0.33 m); Packed with Nova inert i.e. porous silica packing, inoculated with activated sludge | Model assumptions: plug flow of gas phase, no mass transfer resistance, rectangular geometry for biofilm, methanol and DMS are the only growth substrates for microbes, microbial species degrading both substrates are the same; the model incorporates competitive and activation functions of methanol on DMS biofiltration and sets the basis for the description and modeling of VOC mixtures, and successfully predicted the feeding strategies for enhanced biofiltration performance | [139] |
| Styrene | Glass biofilter ($d_{in}$ = 0.005 m, H = 0.41 m) packed with coconut coir inoculated with *Pseudomonas putida* | Model assumptions include: negligible internal and external mass transfer resistance, plug flow bioreactor, uniform distribution of microorganisms throughout the biofilter packing, equilibrium at gas–liquid/liquid–solid interfaces; a deterministic model based on the kinetic parameters is proposed for describing biofiltration behavior for styrene abatement | [163] |

**Table 5.** *Cont.*

| Target Compound (s) | Biofilter Parameters | Model Description | Reference |
|---|---|---|---|
| $CO_2$ as a model compound for absorption studies | Typical packing for biofilters and biotrickling filters, including: porous ceramic rings, Pall rings, polyurethane foam, Lava rock, compost mixture | The paper presents an approach for determining and setting the reference values of mass transfer coefficients for various packing materials used in biofiltration/biotrickling filtration; the approach and obtained values are useful for modeling purposes for a broad range of biofilter configurations; this is an important contribution to further studies of biofiltration modeling | [164,165] |
| Ammonia and hydrogen sulfide | Acrylic tube ($d_{in}$ = 0.1 m; H = 0.61 m); polypropylene, polystyrene or PVC packing elements | The model combines the models described in [153,154]; the model describes co- and counter -current biotrickling filtration for the simultaneous treatment of gas and liquid contaminants; the model assumptions include a plug flow, no oxygen or nutrient limitation, constant biofilm density, substrate concentration at the liquid-biofilm interface is the same as in the bulk liquid, and adsorption, as well as direct transfer of substrate to the biofilm, are insignificant | [143] |
| Methane | Feasibility study—no single experimental verification of the model | Modeling of biotrickling filtration using methanotrophic bacteria; a general, theoretical model for the estimation of methane removal is proposed; the assumptions of model are taken from [155] | [166] |
| Ethyl acetate, toluene (separately) and their mixture | Biofilter with intermittently supplied liquid solution; biofilter ($d_{in}$ = 0.136 m, H = 0.95 m) packed with fibrous peat inoculated with pre-acclimated activated sludge | A dynamic model with the following assumptions: plug flow of gas phase, mass transfer is limited by the diffusion resistance in the liquid phase, adsorption phenomena are negligible, biofilm develops only on the surface of packing elements, inhibition in biodegradation is accounted for; cross-inhibition of ethyl acetate on the toluene removal is confirmed | [140] |
| Ethylene | Theoretical approach | Modeling with the use of genetic algorithms; inverse modeling is used to estimate values of biofiltration parameters | [167] |
| TRS (total reduced sulfur) | Theoretical approach | Sensitivity analysis on the mathematical model for biotrickling filtration of TRS revealed that biofilm properties and kinetics had the most important influence on the model behavior | [168] |

**Table 5.** *Cont.*

| Target Compound (s) | Biofilter Parameters | Model Description | Reference |
|---|---|---|---|
| Hexane | PVC column ($d_{in}$ = 0.083 m, H = 0.53 m); abiotic conditions were tested for determining mass transfer from gas to liquid phase | Modeling resulting in an isomorphous equation describing the maximum fraction of VOC that can be transferred from the gas to liquid phase within the comparison of three types of bioreactors; the biotrickling filter working as a two-liquid phase system exhibited the highest mass transfer performance | [141] |
| Hydrogen sulfide | Pilot scale biotrickling filter ($d_{in}$ = 0.3 m, H = 1.1 m) packed with polyurethane foam cubes inoculated with WWTP activated sludge | A model describing biomass accumulation (the production of solids) related to the pressure head loss increase due to the occurrence of solids by sulfur transformations; the growth rate of biomass, measurements of pressure drops, biomass increase, effluent COD, and total solids were successfully modeled | [169] |
| Nitric oxide | A plexiglass cylinder ($d_{in}$ = 0.08 m, H = 0.3 m) packed with porous ceramic particles inoculated with *Chelatococcus daeguensis* TAD 1 | The model incorporates gaseous chemical oxidation for NO removal during biotrickling filtration, which is novel in biotrickling filtration modeling; the assumptions include plug glow of a gas phase, planar geometry of the biofilm, NO as a single rate-limiting substrate, no excessive biomass and uniform distribution of the biofilm; Monod-type kinetics and mass transfer in the gas–biofilm interface are exploited | [170] |
| Ethyl acetate | Plexiglass column ($d_{in}$ = 0.062 m, H = 1 m) packed with walnut shell, inoculated with *Pseudomonas putida* PTCC 1694 | Experimental model based on zeroth order kinetics with diffusion limitation, assuming gas plug flow, constant biofilm thickness, and diffusion following Fick's law; paper presents optimal results of various model results for the experimental data | [171] |
| Benzene, toluene, ethylbenzene, o-xylene (BTEX) | Perspex pipe ($d_{in}$ = 0.14 m, H = 0.6 m) packed with corn-cobs, inoculated with *Bacillus sphaericus* | CDR (convection–diffusion reactor) model was used to generate VOC concentration profiles along the biofilter | [172] |
| Hydrogen sulfide | A theoretical model | A dynamic model as an extension of a differential reactor model by Kim and Deshusses [138]; the model includes mass balances in gas, liquid, wetted, and non-wetted biofilm; biotrickling filtration performance is mostly affected by $H_2S$ concentration and gas flow rate | [144] |

**Table 5.** *Cont.*

| Target Compound (s) | Biofilter Parameters | Model Description | Reference |
|---|---|---|---|
| Isopropanol | Methacrylate column ($d_{in}$ = 0.144 m, H = 1.2 m); two packing materials were compared: random and structured polymeric media | A model aimed at describing mass transfer properties in biotrickling filters for a hydrophilic compound, including the mass transfer of oxygen, which is a limiting factor in such conditions; the results showed that global mass transfer coefficients describes the mass transfer phenomena in biotrickling filters with the highest accuracy | [146] |
| Hydrogen sulfide | Differential biotrickling filter | A dynamic model for describing the biotrickling filtration of high loads of $H_2S$; assumptions of the model include: biofilm is completely covered with a liquid, adsorption phenomena are negligible, biofilm is uniformly distributed over packing elements, biomass contains active and non-active parts; gas–liquid mass transfer is described by the gas–liquid mass transfer coefficient, the biodegradation kinetics of hydrogen sulfide follow the Haldane equation; the model reveals a high dependence of $H_2S$ biofiltration on oxygen mass transfer | [173] |
| Hydrophilic VOCs: ethanol, ethyl acetate, 1-ethoxy-2-propanol | Two systems were investigated: a laboratory biotrickling filter ($d_{in}$ = 0.144 m, H = 1 m) and an industrial biotrickling filter (packing volume: 49 $m^3$) filled with polypropylene rings inoculated with activated sludge | A dynamic model for biotrickling filtration with intermittent trickling patterns; the liquid phase is considered mobile during spraying episodes and considered stagnant for non-spraying periods; it makes similar assumptions to [173], except for: biodegradation kinetics follow a Monod equation, the presence of biomass in the biofilter increases the resistance for mass transfer from the gas to liquid phase, oxygen limitation is assumed; the model predicts performance, outlet emission peaks, and decreases in concentrations of outlet gases after trickling stops | [103] |
| Methane | PVC column ($d_{in}$ = 0.153 m, H = 1.2 m) packed with polyethylene rings inoculated with methanotrophic bacteria, i.e., *Methylomicrobium album* (ATCC 33003) and *Methylocystis sp.* (ATCC 49242) | A dynamic mathematical model based on mass transfer and kinetic parameters; the model assumptions include: plug flow of gas phase with no axial dispersion, gas–biofilm equilibrium follows Henry's law, diffusion is described by Fick's law, biofilm develops only on the outer surface of packing elements, the surface properties of the biofilm are the same as those for water, uniform distribution and no accumulation of biomass, no adsorption of the pollutant | [174] |

**Table 5.** *Cont.*

| Target Compound (s) | Biofilter Parameters | Model Description | Reference |
|---|---|---|---|
| Styrene | Laboratory ($d_{in}$ = 0.144 m, H = 1.23 m) and industrial biofilters (volume of packing: 0.6 m$^3$), packed with polypropylene rings and structured packing, respectively | A three-phase dynamic model accounting for the convection, mass transfer, diffusion, and biodegradation phenomena of a biotrickling filtration process; the model is based on mass balances of the above-listed phenomena; the model assumes gas phase plug flow, a negligible share of adsorption and axial dispersion pertaining to biofiltration performance, that biodegradation takes place only within the biofilm; a numerical solution to the model by Matlab software; a limiting factor for the biotrickling filtration of hydrophobic compounds is their water solubility | [69] |
| Styrene | Pilot-scale stainless-steel biotrickling filter ($d_{in}$ = 1.084 m, H = 3.51 m) packed with polypropylene Ralu rings inoculated with microorganisms (E-93486 strain) | Simple one substrate model was proposed; the paper presents verification of a two-substrate and one-substrate model for styrene removal from air; oxygen is not a limiting factor for the biotreatment of styrene-polluted air; the simple one-substrate model gave the lowest values of mean relative error, and can be used for predicting the performance of styrene biotrickling filtration | [145] |
| Hydrogen sulfide | A full-scale SULPHUS system was applied using a structured packing media ($d_{in}$ = 3m, H = 5 m) | The paper revises and restates selected models for biodegradation kinetics in biofilms, as well as proposes novel assumptions to be included in the biofilter models; a broad set of data is fit to the existing models, and the most appropriate models for various ranges of $H_2S$ inlet concentrations are proposed | [147] |
| Biogas upgrading | PVC column ($d_{in}$ = 0.03 m, H = 1 m) packed with open pore polyurethane foam inoculated with hydrogenotrophic methanogens (*Methanospirillum hungatei* and additional biomass suspension) | The developed conceptual model includes the transport of hydrogen from the gas to wetted and non-wetted biofilm fractions; the model assumptions include: the uniform spread and thickness of biofilm, biofilm thickness remains constant and cannot be not fully wetted, dynamic variations in biofilm and adsorption on packing material are neglected, diffusion in the biofilm is described by Fick's law, axial dispersion is considered (e.g., due to high substrate concentration), the most sensitive parameters of the model are the gas flow rate, specific surface area, biofilm thickness, and maximum reaction rate; the model can be used for the optimization of biotrickling filters for biogas upgrading, | [172] |

**Table 5.** *Cont.*

| Target Compound (s) | Biofilter Parameters | Model Description | Reference |
|---|---|---|---|
| Toluene and methanol | Glass column ($d_{in}$ = 0.065 m, H = 0.8 m) packed with a mixture of pumice grains and HDPE pall rings inoculated with a consortium of microorganisms, including *Pandoraea pnomenusa* DSM 16536 and *Ralstonia eutrohpa* PTCC1615 | Dynamic model based on mass transfer through gas, liquid, and biofilm phases is developed; assumptions include neglected physical adsorption, plug flow for gas phase, no axial dispersion, planar biofilm geometry, neglected methanol stripping from liquid, and no oxygen limitation; the model was calibrated and validated; the model indicated the fractions of a biofilm / biotrickling filter that are active in the degradation of either toluene or methanol; sensitivity analyses were performed; model can predict the dynamics of VOC biotrickling filtration, together with the determination of kinetic constants | [142] |
| Hydrogen sulfide (biogas upgrading) | A laboratory-scale biotrickling filter ($d_{in}$ = 0.0714 m, H = 0.7 m) packed with polypropylene Pall rings, inoculated with aerobic sludge | Liquid–gas mass transfer correlations are provided; a dynamic model for predicting physio-chemical and biological process during biogas desulfurization using a biotrickling filter; the model is a three-phase model that can describe biotrickling filter operation for variable $H_2S$ inlet loading | [173] |

COD—Chemical Oxygen Demand.

## 5. Concluding Remarks and Future Perspectives

Recent research results with a focus on process modeling for the treatment of air polluted with VOCs in biotrickling filters was presented and discussed. Additionally, due to the historical perspective on the development of biotrickling filtration and its modeling, a brief overview of models for conventional biofiltration was also presented. In the cases of both biological treatment methods mentioned above, modeling started with the description of process performance under steady-state conditions. However, due to dynamic changes in both waste gas flow rates and pollutant concentrations frequently faced in real-life applications, dynamic models for transient conditions were developed. Along with the development of biofiltration modeling, model assumptions and objectives were adjusted to the possible real/industrial conditions, including variable concentrations and compositions of a gas phase component, intermittent spraying patterns, non-uniform growth and distribution of biomass, and non-uniform wetting of a packing/biofilm. These developments resulted in the formulation of various mathematical expressions, which were usually complex and problem-specific, that can aid process design and optimization. However, besides considerable progress in the description of biotrickling filtration mechanisms, the challenges listed in 2005 by Devinny and Ramesh [108] still remain. These include a need for further studies on the effects of substrate interactions when treating VOC mixtures, the effects of substrate types on biodegradation in the biotrickling filter, and a need to predict and model biofilm shape and physico-chemical properties. Nevertheless, no universal model of biofiltration, both for conventional and biotrickling filtration, is available.

Current research on biotrickling filtration concentrates on the means of enhancing process performance, especially with respect to the treatment of hydrophobic volatile organic compounds. As indicated by San-Valero et al. [69], industrial biotrickling filters can present different performance behaviors from laboratory-scale systems, which are mainly due to fluctuating conditions with respect to gas flow rate, shut down periods, and variable inlet concentrations. This statement underlines the special need for models

suitable for dynamic conditions during waste air biofiltration. Hopefully, the overview of recent research shows the increasing number of papers on studies of real waste gas streams or model mixtures containing multiple VOC mixtures that mimic real streams. The application of biotrickling filtration towards biogas upgrading is also rising in importance. This direction of studies and process development seems to be of special importance, and especially consider the current economic and energetic crisis in Europe and in the world.

The future of investigations on biotrickling filtration should still be concentrated on the scaling up process, and offer more industrial applications that produce more results of practical meaning. This is especially important from the perspective of process modeling, which need more general and simplified models for direct industrial applications. In light of sustainability requirements and circular economic interests, combining BTF processes with other processes, including those treating waste with waste, seems to be an unavoidable direction of development. New perspectives are open in terms of managing the by-products of biofiltration with the possibility for energy production, e.g., using microbial fuel cells and the utilization of waste streams from other processes, such as biofilter packing materials or green chemical additives to aid the performance of biofilters. More experimental data or pilot-scale tests are also needed for models that become valuable for practical industrial applications.

**Funding:** This research received no external funding.

**Data Availability Statement:** Data are available in a publicly accessible repository as well as in a publicly accessible repository that does not issue DOIs. All data referred to in this paper are accessible according to the list of references.

**Conflicts of Interest:** The author declares no conflict of interest.

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
