# Peer review of "Removal of Volatile Organic Compounds (VOCs) from Air: Focus on Biotrickling Filtration and Process Modeling"

_processes, doi:10.3390/pr10122531_

Round 1
Reviewer 1 Report
This manuscript is a good contribution to the field of BTF, however the author collectively reviewed different types of bioreactors (bio-filters, BF) and (bio-trickling filter, BTF modeling. BTF and BF are different in many aspects. Furthermore, the contents of the manuscript are not consistent with the tittle of the manuscript "Bio-trickling filtration of air polluted with VOCs: focus on process modelling".
I suggest to revise the manuscript and re-submit taking into consideration of the following points.
1. The title need to be changed. I suggest "Bio-trickling filtration for removal of air polluted with VOCs: focus on process modelling"
2. I suggest the author to focus more on bio-trickling filter modeling, rather than mixing both BF (bio-filter) and BTF. If you want to consider both BF and BTF, then the title is not consistent with the content.
3. BF and BTF are different. Sometimes in the literature the term "bio-filtration" can be misleading. For example, BF does not require continuous nutrient recirculation as compared to BTF. If the author decides to include both BF and BTF, then the title need to be changed , more references and separate sections for BF and BTF need to be included.
4. Only 3 keywords are given. Please add more key words.
5. Line 41. Objective is to focus on BTF but the contents include both BF and BTF. The objective may need to be modified to match the contents of the manuscript.
6. Line 48. "Bio-trickling filtration, a type of a group of processes termed collectively as bio-filtration...". I disagree with this statement.
7.Line 109. "bio-filtration performance can be regulated by manipulating the trickling liquid velocity and frequency". I disagree with this statement. This is only true for BTF. In BF there is no recirculation of nutrients.
8. Tables 1-4 data are mixed. I suggest the author to separate BF related work from BTF.
9. Conclusion section. Line 474. "due to the historical perspective on the development of bio-trickling filtration, a brief overview of conventional biofiltration is also presented". However, you have mixed both BF/BTF and listed in the tables 1-4 and discussed throughout.
10. Some references are not needed, may be removed. For example, references 3,4, 5...etc.
11. Avoid reference from conferences (Ref. 22) unless you can provide a link or DOI for this.
12. Reference 87. Correct the word "bioprocess".
13. Reference 104. Please provide the original link rather than from Research Gate website
14. Reference 134. Not complete.
15. Some important references on BTF are missing, for example, “Dynamic modeling and analysis of bio-trickling filters in continuous operation for H2S removal" https://doi.org/10.1007/s10098-013-0697-0
Author Response
Please see the attachement

Reviewer 2 Report
Comments to the manuscript processes-2009992: Biotrickling filtration of air polluted with VOCs. Focus on process modelling
The present review deals with degradation of both single and mixed VOC conditions by BTFs and BFs and shows a good overview about experimental and mathematical data on the performance / expected performance of these biofilter systems. The review shows a good overview over literature, even though some aspects are not taken into account (most probably due to the length of the manuscript). However, they should be addressed (see below). The manuscript is also of high quality according to lingual aspects and relevance of these data for public; therefore this manuscript should be published after considering the following aspects:
· Line 14: …allows deeper understanding of…
· Line 33: established instead of practiced
· Line 34: for a review listing of alternative waste air treatment methodologies should also cover techniques like condensation, membrane processes, oxidative catalysis, UV-oxidation, non-thermal plasma, bioscrubber, biotrickling filter, biofilter and combinations of techniques. Potential references might be:
o Condensation:
§ https://doi.org/10.1016/S0009-2509(02)00158-6
§ https://doi.org/10.3390/pr9091658
o Membrane processes:
§ US Department of Energy, 2001; DOE/EM-0614
§ https://doi.org/10.1016/0376-7388(96)00145-7
o Oxidative catalysis:
§ https://doir.org/10.1016/S1872-2067(15)61007-5.
§ https://doi.org/10.1016/j.jclepro.2019.04.258
o Non-thermal plasma:
§ https://doi.org/10.3390/su12219240 (combined BTF and NTP)
§ https://doi.org/10.1016/j.cej.2015.01.055
o UV-oxidation:
§ https://doi.org/10.1016/j.jhazmat.2011.06.092 (combined with BTF)
§ https://doi.org/10.1016/j.cej.2015.04.016 (combined with BTF)
§ https://doi.org/10.1016/j.biortech.2009.05.074 (Combined with BF)
§ https://doi.org/10.1080/09593330.2015.1068375 (combined with
· Line 41: update instead of up-date
· Line 48-49: …biofiltration, enables removal of gas contaminants….
· Line 60. …the treated air leaving the biotricklingfilter contains only …
· Line 96: The inlet load presented here is a specific inlet load as it is referred to the volume of the package material.
· Line 103: Beside inlet concentration EBRT as contact time should also be presented. This aspect should be added.
· Line 110: It should be taken into account that frequency of trickling process cannot be implemented in all applications as agricultural applications in detail strongly require continuous irrigation.
· Table 1: Line Packing material: Please explain what is meant by bio-based packing materials, especially in the focus of biotrickling filters.
· Line 156 and Figure 1 (also Line 172): It should be mentioned that degradation of hydrophobic VOCs can also be accelerated by use of package materials with enhanced adsorptive capacity, i.e. hydrophobic VOCs can adsorb and accumulate on these carriers and mass transfer to cell and final biodegradation is enhanced by higher VOC gradient between carrier and biofilm. Adequate papers dealing with this aspect are as examples:
o https://doi.org/10.1002/jctb.3779 using Bio-airSPHERES
o https://doi.org/10.1016/j.cej.2006.06.023 using synthetic fiber strings
o Park et al. 2008, Envrion. Eng. Res. Vol 13, No. 1, pp. 10-27 using inorganic/polymeric composite Bio-M
o https://doi.org/10.1016/j.cej.2018.08.140 using Bio-airSPHERES
o https://doi.org/10.1016/j.chemosphere.2020.127093 using foam ceramic composite filler
· Line 175: …great majority between 30-100s while…
· Line 177: …range from about several to thousands of… sentence is incomplete; most probably several hundred
· Table 3: First letter of each cell should be all the time a large or a small letter, but not mixed.
· Line 229: The statement that conventional biofiltration is well described in literature should be underlined by references.
· Line 232 & 234 & 340: Authors à authors
· Line 235: Eliminate one ‘that’
· Line 275: Are specific parameters like De, K0 and K1 presented in the reference or where can you get them?
· Line 285-287: What is the function of a pure liquid mass transfer model? Does it include somehome variables or terms to describe biodegradation in the biofilm? Please add this information.
· Line 308: Most probably acidification took place. Was it solely caused by CO2 uptake or also by partial oxidation of ethanol to acetic acid. Please add this information.
· Line 316: Which parameter affect b1? Please add them and give a critical view of these parameters.
· Line 356: …influences of …. were investigated and modeled.
· Line 401: Why does oxygen limitation gets a problem in case of BTFs but not in case of BFs?
· Line 409-410: Adsorption seems to be only relevant for rare package materials as main materials are made of plastic with low adsorption capacity. Might be this aspect should be mentioned.
· Line 456: Reader à reader
· Line 434-466: Is there no aspect of microbial species or induction level as relevant parameter to assume biodegradation kinetics in this models? If not, why do the omit this aspect?
Author Response
Please see the attachement

Reviewer 3 Report
Please find the attached file for needed revisions

Round 2
Reviewer 1 Report
Please see the attachment.
